# Porous Polydimethylsiloxane Elastomer Hybrid with Zinc Oxide Nanowire for Wearable, Wide-Range, and Low Detection Limit Capacitive Pressure Sensor

**DOI:** 10.3390/nano12020256

**Published:** 2022-01-14

**Authors:** Gen-Wen Hsieh, Liang-Cheng Shih, Pei-Yuan Chen

**Affiliations:** 1Institute of Lighting and Energy Photonics, College of Photonics, National Yang Ming Chiao Tung University, 301, Section 2, Gaofa 3rd Road, Guiren District, Tainan 71150, Taiwan; 2Institute of Photonic System, College of Photonics, National Yang Ming Chiao Tung University, 301, Gaofa 3rd Road, Section 2, Guiren District, Tainan 71150, Taiwan; lighttime0625@yahoo.com.tw (L.-C.S.); aazz55255361@gmail.com (P.-Y.C.)

**Keywords:** capacitive pressure sensor, porous polydimethylsiloxane, stress-sensitive, wearable electronic, zinc oxide nanowire

## Abstract

We propose a flexible capacitive pressure sensor that utilizes porous polydimethylsiloxane elastomer with zinc oxide nanowire as nanocomposite dielectric layer via a simple porogen-assisted process. With the incorporation of nanowires into the porous elastomer, our capacitive pressure sensor is not only highly responsive to subtle stimuli but vigorously so to gentle touch and verbal stimulation from 0 to 50 kPa. The fabricated zinc oxide nanowire–porous polydimethylsiloxane sensor exhibits superior sensitivity of 0.717 kPa^−1^, 0.360 kPa^−1^, and 0.200 kPa^−1^ at the pressure regimes of 0–50 Pa, 50–1000 Pa, and 1000–3000 Pa, respectively, presenting an approximate enhancement by 21−100 times when compared to that of a flat polydimethylsiloxane device. The nanocomposite dielectric layer also reveals an ultralow detection limit of 1.0 Pa, good stability, and durability after 4000 loading–unloading cycles, making it capable of perception of various human motions, such as finger bending, calligraphy writing, throat vibration, and airflow blowing. A proof-of-concept trial in hydrostatic water pressure sensing has been demonstrated with the proposed sensors, which can detect tiny changes in water pressure and may be helpful for underwater sensing research. This work brings out the efficacy of constructing wearable capacitive pressure sensors based on a porous dielectric hybrid with stress-sensitive nanostructures, providing wide prospective applications in wearable electronics, health monitoring, and smart artificial robotics/prosthetics.

## 1. Introduction

To meet the growing demand for wearable healthcare electronics and human–machine interfaces, nanocomposite materials that employ flexible polymers in conjunction with stimuli-sensitive nanostructures have engrossed significant attention for the consciousness of temperature [1,2,3,4,5], moisture [6,7], light [8,9], and touch [10,11,12,13]. Unlike conventional composites or a sole type of material, polymer-based nanocomposites may offer considerable enhancement in thermal, electrical, optical, chemical, or mechanical properties. Among them, a variety of nanostructures with highly conductive [14,15,16], dielectric [17,18,19], piezoelectric [20,21,22], triboelectric [23,24,25], photo-responsive [26,27,28], or stress-sensitive [10,29,30,31] properties have been studied for flexible pressure sensors because of their potential in amplifying stress/strain sensing capability for human motion detection, health diagnosis, and electronic skin.

Recently, capacitive pressure sensing has been intensively investigated, owing to its simple geometry, low power consumption, and good environmental stability [18]. In a simple parallel plate capacitor, capacitance is commonly changed as a function of external pressure. In particular, elastomeric polydimethylsiloxane (PDMS)-based silicone rubbers have been chosen as the dielectric layer for capacitive pressure sensors due to their superior flexibility, nontoxicity, and low material cost [32,33]. However, restricted by their viscoelastic property and low compressibility, such PDMS films are not able to produce enough deformation upon very small pressures. Moreover, after pressure unloading, their recovery time of return to initial condition is rather slow [18]. To overcome these limitations, adoption of micro-nano-scaled structures or pores into an elastomeric matrix is a possible means to improve sensing performance [34,35,36,37,38]. With applied pressure, these embedded air gaps or pores in the deformed PDMS films can induce massive volumetric deformation as well as increments in effective dielectric permittivity, which can in turn increase the capacitance change and pressure sensitivity. However, such enhancements can only be achieved in the low-pressure regime; when these air gaps or pores are nearly squeezed under heavy load, the flattened PDMS becomes hardly compressible. Additionally, the fabrication processes for these flexible pressure sensors are complicated, high-cost, and difficult to control, which is not practical for human tactile interactions.

In this regard, an appropriate way to generate high-performance capacitive pressure sensors is to tailor low-dimensional nanostructures with porous or microstructured elastomeric polymers by merging their functionalities. Owing to the high surface area to volume ratio, several types of nanostructures can promote the formation of a large interfacial area between polymer and nanofiller, which is beneficial to a higher level of polarization. For instance, Mu et al. demonstrated a nanocomposite dielectric layer of CaCu_3_Ti_4_O_12_ ceramic nanocrystals (with high dielectric permittivity) and porous PDMS matrix (with low dielectric permittivity) [39], presenting low compressibility and high sensitivity compared to that of pure PDMS. Pruvost et al. produced a composite dielectric foam decorated with conductive carbon black particles on the inner surface of pores [40], which can enhance the effective dielectric permittivity and the capacitance change under applied stress. Moreover, Kou et al. proposed a wireless flexible pressure sensor containing a dielectric layer of graphene/PDMS sponge sandwiched with patterned Cu antenna and electrode [41]; the air holes between the graphene particles (like numerous parallel mini-capacitors) are very sensitive to deformation. However, utilizing zero-dimensional nanoparticles or two-dimensional nanoflakes in the polymer matrix may likely cause the problem of aggregation and uneven dispersion. These shortcomings have prompted the pursuit of augmentative dielectric nanocomposites in the quest for high-performance pressure sensors. According to our understanding, there is no previous study based on porous polymer-based nanocomposite hybrids with one-dimensional nanostructures for capacitive pressure sensing applications.

Here, we propose a novel nanocomposite dielectric material where one-dimensional ZnO nanowire is incorporated into a porous elastomeric polymer for the formation of flexible capacitive pressure sensors. ZnO nanowire is a unique high-aspect-ratio, biocompatible, and low-cost material that exhibits semiconducting, piezoelectric, and pyroelectric multiple properties and that can be easily dispersed in water/organic solvent and polymer matrix. The porous polydimethylsiloxane (PDMS)-based nanocomposite of closed porosity is prepared by using a porogen-assisted process, and ZnO nanowires are randomly distributed in PDMS. Upon external compression, a large change in dielectric permittivity consolidating massive variations in capacitance can be achieved by enhancing the interfacial polarization and elastic modulus of the nanocomposite. The measured pressure sensitivity of the fabricated capacitive pressure sensors shows a remarkable improvement of more than 21 times when compared to that of flat PDMS devices; it can distinguish a subtle pressure of ~1.0 Pa with an ultrafine resolution as low as 0.4 Pa. The nanocomposite dielectric layer also reveals good stability and durability after 4000 loading–unloading cycles and a wide detection range, showing a great potential for the perception of various human motions.

## 2. Experimental

### 2.1. Growth of ZnO Nanowires

ZnO nanowires were produced by vapor phase carbothermic reduction [10,42,43]. A mixture of ZnO powder (99.9%, Alfa Aesar, Ward Hill , MA, USA) and graphite (Sigma-Aldrich, Burlington, MA, USA) (weight ratio ~1:1) in an alumina boat was loaded into the center of a quartz tube furnace, and a Si <111> substrate with a gold film (1.0 nm thick) was placed at the downstream end. Then, the temperature of the tube center was increased to 950 °C under an Ar/O_2_ flow (50/1 sccm; total pressure: 2 mbar) to promote the nanowire growth.

### 2.2. Fabrication of ZnO Nanowire–Porous PDMS Capacitive Pressure Sensors

Prior to incorporation of nanowire into PDMS, as-grown ZnO nanowires on the Si substrate were dispersed into ethanol by sonication. The nanowires of different weight ratios (0, 0.5, 1.0, and 1.5 wt%, respectively) were blended with a proper amount of PDMS prepolymer (Sylgard^®^ 184A, Dow Corning) by vigorous stirring to form a thorough viscous solution. After ethanol removing, the prepolymer blend was mixed with a porogen solution containing water/2-propanol (volume ratio of 3:1) at 3600 rpm for 30 min for the formation of porous elastomers [44]. The solution of pure PDMS or ZnO nanowire–PDMS prepolymers with porogen was further mixed with a curing agent (Sylgard^®^ 184B) (weight ratio ~10:1) at 2000 rpm for 60 min in order to produce well-dispersed microdroplets in the PDMS-based solution. 

Further, the proposed capacitors were fabricated on a transparent film of polyethylene terephthalate (PET, Nan Ya Plastics, thickness ~100 μm). Bottom electrodes of poly (3,4-ethylenedioxythiophene):poly(styrenesulfonate) (PEDOT:PSS, Clevios™ PH 1000) polymer (thickness: ~5 μm) were ink-jet printed by a commercial printer (Epson L120). Thus, the ZnO nanowire–PDMS prepolymer solution containing microdroplets of water could be blade-casted onto the PET with a subsequent baking at 70 °C for 24 h in order to achieve the thermal curing and solidification of the PDMS. In the meantime, the porogen microdroplets (water/2-propanol) confined in the PDMS could completely evaporate and permeate through PDMS at the curing temperature. Note that the water/2-propanol (volume ratio of 3:1) porogen provides an increased distribution of microdroplets inside PDMS, when compared with the porous network obtained using water alone as porogen.

Finally, a fully polymerized ZnO nanowire–porous PDMS film (thickness of ~200 μm) could be generated. The top PEDOT:PSS electrodes were deposited on the top of the dielectric layer, forming a sandwich-like capacitor layout (area: 10 mm × 10 mm). As a comparison, pristine porous PDMS (without nanowires) and flat PDMS (without pores and nanowires) were prepared at the same time via a similar manner. Note that the flat PDMS film (~200 μm) on a PEDOT:PSS/PET substrate was formed by casting its prepolymer, mixed with curing agent at a ratio of 10:1 (w/w), degassing in a vacuum oven for 30 min, and curing in the oven at 70 °C for 4−6 h. 

### 2.3. Characterization and Measurement

The morphology and crystalline structure of the ZnO nanowires were characterized by high-resolution transmission electron microscopy (HRTEM, JEM-2100F CS STEM, JEOL, Akishima, Japan) and X-ray diffraction (D8 DISCOVER Plus-TXS, Bruker, Billerica, MA, USA). The cross-sectional morphology of flat PDMS, porous PDMS, and ZnO nanowire–porous PDMS nanocomposites was analyzed by scanning electron microscopy (SEM, SU8000 FESEM, HITACHI, Tegama, Japan). Capacitance vs. pressure loading measurements were measured on Agilent E4980AL Precision Impedance Analyze (at 860 kHz frequency with A.C. bias of 2.0 V). The relative dielectric permittivity was measured using a dielectric test fixture (Agilent 16451B, Keysight, Santa Rosa, CA, USA). The pressure sensing performance of the ZnO nanowire–porous PDMS was presented alongside those of the flat PDMS and pristine porous PDMS devices. All experiments were performed at room temperature.

## 3. Results and Discussion

A schematic flow for the fabrication of the proposed flexible capacitive pressure sensors based on a nanocomposite dielectric layer of ZnO nanowire and porous PDMS elastomer is illustrated in Figure 1. These pressure sensors were fabricated on PET substrates with PEDOT:PSS conducting electrodes (thickness: ~5 μm). Four porous composites were studied with 0, 0.5, 1.0, and 1.5 wt% of nanowires, respectively. The nanocomposite dielectric layer of ZnO nanowire–porous PDMS nanocomposite (thickness: ~200 μm) was formed by blade-casting. The incorporated wurtzite ZnO nanowires were ~30−70 nm in diameter and ~5−7 μm in length and revealed crystalline characteristics throughout by SEM, selected area electron diffraction (see Figure 2a), and X-ray diffraction. The HRTEM image displays (0002) lattice fringes with interplanar spacings of ∼2.6 Å (Figure 2b). As can be seen from Figure 2c,d, both porous PDMS and ZnO nanowire–porous PDMS elastomers exhibit similar porous morphology, where the pore sizes were observed to be 4.2 ± 1.8 μm (see the Appendix A) with a calculated porosity of ~30.0%. Note that the porosity of the PDMS film can be adjusted through the mixing of porogen solution and PDMS polymer with different weight ratios. The cross-sectional SEM image of the nanocomposite films also proved that those nanowires (indicated by arrow marks) were distributed uniformly in the PDMS matrix without interrupting the porous networks. Further, the visual appearance of the nanocomposite film was clear and uniform; the ATR-FTIR (attenuated total reflection-Fourier transform infrared) spectra of porous PDMS and ZnO nanowire–porous PDMS were almost identical (Appendix A). Digital photographs of the fabricated capacitive pressure sensors are also shown in Figure 2e.

Subsequently, we explored the pressure sensing capabilities of these porous capacitive pressure sensors with different loading of ZnO nanowires. To ensure the whole sensor was receiving uniform pressure load, a small plastic open container (contact area: ~10 mm × 10 mm, base pressure: ~10 Pa) was firmly attached to the sensor surface. A fixed amount of water droplets (20 μL) was carefully dispensed drop by drop into the container by using a micropipette (equivalent to a subtle pressure of ~2 Pa). The measured capacitance value of the sensor is denoted as *C*_0_ (without pressure load) and *C* (with pressure load), respectively; the value can be determined based on *C* = *ε_r_ε*_0_*A*/*d*, where *ε_r_* and *ε*_0_ are relative permittivity of the dielectric layer and permittivity of vacuum, respectively; *d* is the separation between two electrodes; and *A* is the area of the overlapped electrodes [45]. 

Figure 3a shows the relative capacitance change (Δ*C/C*_0_) as a function of the applied pressure (*P*) for the capacitive pressure sensors with different dielectric layers, where Δ*C* represents the change between *C*_0_ and *C*. These curves, presenting how the capacitive pressure sensor senses the external pressure change, clearly undergo a sharp upward trend in the low-pressure regime and a smooth increment in the high-pressure regime. For flat PDMS sensors, the Δ*C/C*_0_ increased linearly with small applied pressure due to the elastic deformation between two electrodes. Further, under moderate-to-large pressure, the Δ*C/C*_0_ was nearly saturated, since the deformed PDMS became hard to deform. For porous PDMS sensors without nanowire loading, the sensor was compressed upon applied pressure, causing the embedded air pores to shrink. Thus, the sensor capacitance increased with the applied pressure under the combined effects of the pore thickness reduction and the dielectric permittivity increase, consequently promoting pressure sensitivity [36,44]. For ZnO nanowire–porous PDMS capacitive pressure sensors specifically, the performance in relative capacitance change was substantially better than those of the flat PDMS and porous PDMS devices. This was probably attributed to the elastic modulus of the nanocomposite dielectric elastomer, which can endow the dielectric film with good buffer effect upon external stress, and the gradual increase in dielectric permittivity during compressing (Appendix A). Consequently, the ZnO nanowire–porous PDMS elastomer can induce much higher variation in dielectric permittivity under the same compression pressure, which can operate properly over a wide range of 0–50 kPa.

By adjusting the nanowire loading, these ZnO nanowire–porous PDMS dielectric films present different pressure sensing capabilities. The relative capacitance changes of the ZnO nanowire (1.0 wt% loading)–porous PDMS nanocomposite were much more significant, which consequently gave the best comprehensive sensing performance (Figure 3b). This was closely related to the volumetric change of the porous structure and embedded nanowire of the nanocomposites during the compression process. The sensitivity of capacitive pressure sensors can be extracted through the slope of the curve of relative capacitive change versus applied pressure, as *S* = Δ *(*Δ*C/C*_0_*)/*Δ*P* [45]. As can be seen, the ZnO nanowire (1.0 wt% loading)–porous PDMS sensor has the highest sensitivity of *S* = 0.717 kPa^−1^ in the regime of the 0–50 Pa range, presenting an improvement of 1.8 and 21.1 times over that of porous PDMS and flat PDMS, respectively. In the regime of 50–1000 Pa, the sensitivity of ZnO nanowire–porous PDMS, porous PDMS, and flat PDMS reduced to 0.360, 0.286, and 0.014 kPa^−1^, respectively. In the regime of 1000–3000 Pa, notably, the ZnO nanowire–porous PDMS nanocomposite device still displayed an impressive pressure sensitivity of *S* = 0.200 kPa^−1^, which was ~100 times greater than that of the flat PDMS sensor and ~1.4 times greater than that of the porous PDMS sensor. In contrast, the values of those porous PDMS and flat PDMS were degraded significantly because of their hardly compressed status. When the applied pressure was larger than 3000 Pa (up to 50 kPa in our test range), the pressure sensitivities of flat PDMS, porous PDMS, and ZnO nanowire–porous PDMS capacitors were <0.001 kPa^−1^, ~0.002 kPa^−1^, and ~0.004 kPa^−1^, respectively. We also noticed that excessive content of ZnO nanowire loading (i.e., >1.0 wt%) may adversely affect the pressure response. For instance, the sensitivity of the porous PDMS-based device containing 1.5 wt% ZnO nanowires was reduced to 0.419 kPa^−1^. This degradation clearly showed that excessive loading of nanowire did not positively contribute to the sensing performance, probably due to the increased elastic modulus of the nanocomposite. This can in turn diminish capacitance response and lead to a poorer sensitivity. Further, our attempts with a ZnO nanowire (2.0 wt% loading)–porous PDMS device did not succeed, since nonuniform nanowire distribution with certain aggregation was observed in the nanocomposite (Appendix A). This trade-off relationship is in line with previous observations in ceramic nanocrystals–PDMS capacitive pressure sensors [39]. Therefore, careful optimization and improvement on the interfacial compatibility of nanofillers and polymer matrix are necessary to maximize the positive contribution of ZnO nanowires. Overall, Figure 3c and Appendix A summarize the measured maximum sensitivities and the mean values of relative capacitance change for the ZnO nanowire (1.0 wt% loading)–porous PDMS, porous PDMS, and flat PDMS capacitive pressure sensors at different applied pressure ranges.

To address the possible sensing mechanism, we anticipate that both air pores and ZnO nanowires incorporated in the elastomeric film provide beneficial effects when the proposed sensor is under compression. Figure 3d graphically depicts the relative capacitance change to the corresponding pressure loading for the sensors with flat PDMS, porous PDMS, and ZnO nanowire–porous PDMS dielectrics, respectively. As a parallel-plate capacitor, the variation of capacitance is determined by the dielectric property of the nanocomposite layer and the separation between two electrodes. Thus, under a uniform external pressure, the capacitance of the deformed flat PDMS film (see i) is simply a function of the reduced thickness of the dielectric film, since its relative permittivity (*ε_r,flat_*) remains constant. Further, the porous PDMS (ii) can produce a larger deformation, since the presence of air pores in PDMS makes the pressure sensing material softer due to the reduced compressive modulus of the dielectric layer. These pore volumes can be gradually shrunk, and the air fraction in PDMS reduces; this can lead to an increase in the effective relative permittivity (Δ*ε_r,pore_*) and further enhance the change rate of capacitance as well as the pressure sensitivity. While the pressure loading is high, the pores will be nearly closed; the relative permittivity of the porous film will reach a saturation value, approaching the value of flat PDMS. Meanwhile, the mechanical property of the stressed porous PDMS film tends towards that of flat PDMS, becoming hardly compressible. For a porous nanocomposite–elastomer hybrid with nanowires (iii), the pressure sensing performance can be determined by comprehensive consideration of the elastic modulus of the dielectric and the change in relative permittivity (*ε_r_*) during deformation. When the pressure loading is low, both the porous PDMS and ZnO nanowire–porous PDMS have similar air pore-induced deformation. This volumetric pore closing can in turn shorten the separation between ZnO nanowires and profoundly enhance the effective relative permittivity (Δ*ε_r,pore+ZnO_*) of the ZnO nanowire–porous PDMS layer, rather than that of porous PDMS. Thus, the total charge capacity and the change in capacitance will enhance intensely, leading to an augmented pressure sensing functionality. When the pressure loading is high, the pores in both films are nearly closed with very little ε_r_ increment. Thus, the sensitivity of the squeezed porous films is rather dominated by their elastic modulus. The flattened porous PDMS becomes hardly compressible; in contrast, the flattened ZnO nanowire–porous PDMS can still retain a certain deformability due to the nanowire-enhanced elastic modulus, contributing to a wider operation pressure range. In addition, we anticipate that the incorporation of ZnO nanowire can change the dielectric properties of the ZnO nanowire–porous PDMS nanocomposite. According to our measurement results, the capacitance (*C*_0_) and relative permittivity (*ε_r_*) of the nanocomposite film (*C*_0_ = 9.80 pF, *ε_r_* = 2.22 for 1.0 wt% nanowire loading) are higher than those of porous PDMS (8.94 pF, 2.02). This could prove that the addition of ZnO nanowire into porous PDMS matrix can enhance the dielectric polarization and relative permittivity due to the interfacial polarization between the ZnO nanowire and porous PDMS matrix. In short, the ZnO nanowire–porous PDMS has a remarkable sensing response compared to that of the pristine porous PDMS and flat PDMS, revealing much higher sensitivity under identical applied pressure.

Moreover, the time-resolved cyclic loading behaviors for a ZnO nanowire (1.0 wt% loading)–porous PDMS capacitive pressure sensor, a porous PDMS sensor, and a flat PDMS device are shown in Figure 4a. For the ZnO nanowire–porous PDMS nanocomposite device, a regular and steady sensing pattern was observed; the capacitance changed rapidly during pressure loading and unloading (applied pressure: 300 Pa). Its response time and recovery time were 260 ms and 400 ms, respectively. In contrast, the porous PDMS and flat PDMS devices both revealed slower response times of 300 ms and 650 ms as well as longer recovery retention times of 520 ms and 780 ms, respectively. Thereby, the incorporation of ZnO nanowires into the porous PDMS matrix can enhance the capacitance change and sensitivity profoundly. Moreover, to examine the limit of detection of the proposed sensor, water droplets of 10 mg per drop were carefully loaded by using a precision micropipette. As shown in Figure 4b, the additions of different droplets were clearly observed; specifically, the step increment of the capacitance change was primarily attributed to the applied water droplets, displaying a subtle pressure of only 1.0 Pa (one drop), 2.0 Pa (two drops), and 3.0 Pa (three drops), respectively. This demonstrates that our proposed sensor is rather responsive to external ultralow load. Meanwhile, we also placed different numbers of sesame seeds on the pressure sensor (Appendix A). The curve shows the progress from putting zero seeds to twelve seeds; the average seed weight is ~4 mg. Starting from three seeds being added, notably, the capacitance response presents a linear and steady increment each time a further seed is placed on the sensor, equating to an ultrafine resolution as low as only ~0.4 Pa. Hence, our device is quite sensitive and seemingly comparable with the limits of detection achieved in other studies: 0.1 Pa [36,37], 1 Pa [31,38], 5 Pa [41], 9 Pa [40], and 20 Pa [30].

In addition, we observed the long-term endurance of the ZnO nanowire–porous PDMS capacitive pressure sensor by allocating it to 4000 cycles of pressure loading–unloading (loading: 0.65 s; unloading: 0.65 s; applied pressure: 300 Pa). As depicted in Figure 4c and Appendix A (for porous PDMS), the device kept very well in pressure sensing without obvious degradation during the test. For instance, mean signals under different cycles, i.e., between the sequence of 100−110, 1990–2000, and 3890−3900 (see insets), were almost the same (at loading: 10.79, 10.79, and 10.73 pF; at unloading: 9.80, 9.69, and 9.67 pF, respectively), presenting a negligible change of 0.6% in loaded capacitance and 1.3% in base capacitance. Therefore, our ZnO nanowire–porous PDMS sensor underwent negligible capacitance variation during the dynamic 4000 cyclic loading processes. It could be concluded that with the incorporation of ZnO nanowire the performance of the sensor is highly repeatable, durable, and stable.

Furthermore, we demonstrate that our proposed capacitive pressure sensor can maintain a reasonable response over a wide dynamic range, from 1.0 Pa to 50 kPa (by placing different objects, i.e., three sesame seeds, a paper clip, a playing card, a lighter, two game controllers, a standard weight, a speaker, a soft drink can, and even up to a ceramic mug). All in all, our nanocomposite pressure sensor can nearly cover the overall tactile pressure range. The recent progress on flexible capacitive pressure sensors, in terms of device components, sensitivity, limit of detection, response time, and operation pressure range, is also summarized in Appendix A.

Based on the superior features of our ZnO nanowire–porous PDMS capacitive pressure sensor, the applications in several physiological activities were further investigated. As shown in Figure 5a, the sequential firm or soft touches given by a Chinese calligraphy apprentice can induce subtle pressure variations; on that account, our sensor can clearly detect and convert the force given to the brush into capacitances. In this case, the mean value of Δ*C* was ~0.95 pF, whereas the mean value for firm touches was ~1.2 pF and was ~0.65 pF for soft touches. Figure 5b shows the real-time capacitance response to an index finger excise. The finger bending and extending (in different bending angles of ~0°, 30°, 45°, 60°, or 90°) can change the stress applied onto the attached capacitor. When the finger was bent at different positions, the relative capacitance change increased accordingly; when the finger was fully straightened, the value then restored to its original state. Besides direct-contact stimuli, we also demonstrated the sensing of non-contact airflow by using an air spray gun. As can be seen in Figure 5c, the discernible signal peaks corresponding to the rapidly applied pulsed airflows can be observed simultaneously, which indicates the potency of adapting our proposed devices for non-contact signal monitoring. Another attempt presented in Figure 5d is for vocal cord movement detection, where a fabricated ZnO nanowire–porous PDMS sensor was placed at the throat region of a volunteer by using adhesive tape. Rapid and obvious responses with different patterns of signals were successfully recorded while speaking “one–two−three−four−five”. Though not optimized yet, these demonstrations express the potential of utilizing our ZnO nanowire–porous PDMS sensor as a wearable device for human motion monitoring. In addition, a prototype 4 × 4 multipixel sensor array based on the proposed nanocomposite dielectric layer was fabricated for spatial pressure distribution tests. The area of each pixel was 5 mm × 5 mm, and the spacing between electrodes was 5 mm. As shown in Figure 5e, four different silicone rubber stamps (shaped with “N”, “C”, “T”, and “U”) were applied to the sensor array. Accordingly, each letter, which was presented in terms of induced capacitance change at the loaded pixels, was legible, and the crosstalk between pixels was rather small. Overall, the prototype sensor array can proceed with spatial pressure mapping, revealing potential for use in human−machine interface applications.

As a specific application demonstration, we have successfully applied our capacitive pressure sensor for water pressure monitoring. As shown in Appendix A, a sensor attached to a supporting glass slide was placed inside a vertical acrylic cylinder (length: 120 cm) filled with water. The device can clearly transduce underwater pressure at different underwater levels; the change in capacitance presents a sharp slope in the range of shallower water levels (0–5 cm) and a gentle slope in the range of deeper water levels (5–100 cm). By this experiment, the ZnO nanowire–porous PDMS pressure sensor is proved to be able to detect tiny changes in water pressure and, thus, may be helpful for underwater pressure sensing research.

In brief, this report displays the novelty of developing ZnO nanowire–porous PDMS nanocomposite dielectrics for wearable, wide-range, and low detection limit capacitive pressure sensors, which have been found to be highly responsive to subtle stimuli but vigorously so to gentle touch and verbal stimulation. For hybrid stimuli-sensitive nanostructure–porous polymer nanocomposites, there is plenty of potential for further enhancement. For instance, finding a novel way (perhaps by using hierarchical self-assembly, femtosecond laser, 3D printing, emulsion-templated phase separation, etc.) to control the size, distribution, and position of the pores in polymer matrix is intriguing; this can influence the sensor performance and reliability for practical applications. Experiments are also being conducted to examine chemical/physical stable materials (such as Ecoflex, polyurethane, polyimide, styrene-ethylene-butylene-styrene, etc.) and optimize device properties and to evaluate the working stability under different environmental conditions (e.g., temperature, humidity, underwater/oil). Additional discussion will center on the effect of interfacial affinity and interfacial polarization of functionalized nanostructures with various dielectric polymers. Overall, the demonstration of the present strategy may serve as a source of inspiration for the development of next-generation flexible pressure sensors.

## 4. Conclusions

To summarize, a novel nanocomposite capacitive pressure sensor was developed by hybridizing zinc oxide nanowire into a porous polydimethylsiloxane dielectric layer. By choosing the appropriate loading of nanowire into the matrix, the maximum sensitivity of the capacitive sensor was 0.717 kPa^−1^ at a subtle pressure range and 0.200 kPa^−1^ at a medium pressure range, exhibiting a broad detection range from as low as a sesame seed up to as high as a ceramic mug and revealing excellent operation stability over 4000 cycles. Furthermore, practical application measurements further showcased that this nanocomposite porous pressure sensor was capable of detecting finger bending, calligraphy writing, voice-induced vibration, air blowing, and hydrostatic underwater pressure, as well as identifying object spatial distribution. The results signify that these flexible nanowire–porous elastomeric nanocomposites could be used in capacitive-based pressure sensors for diverse applications, such as health monitoring and skin-like electronics.

## Figures and Tables

**Figure 1 nanomaterials-12-00256-f001:**
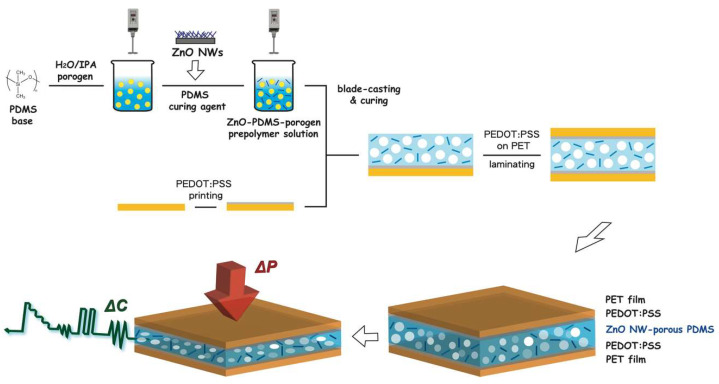
Schematic illustration for fabricating a porous polydimethylsiloxane-based nanocomposite dielectric film hybrid with zinc oxide nanowire for the formation of flexible capacitive pressure sensors.

**Figure 2 nanomaterials-12-00256-f002:**
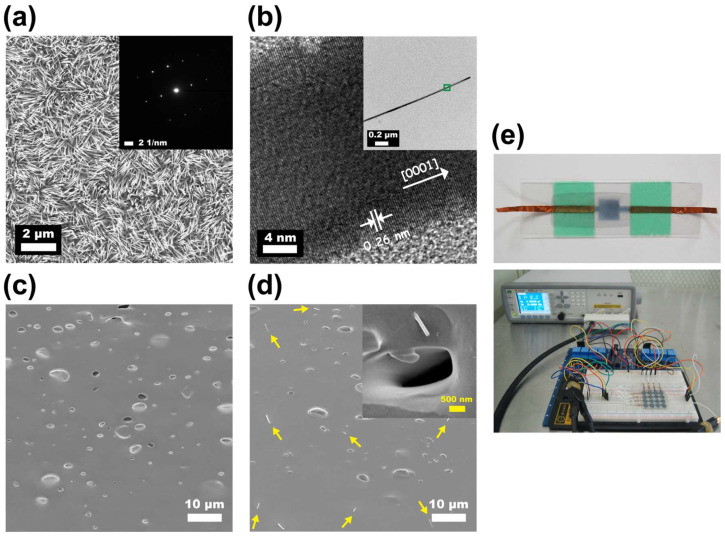
(**a**) SEM image of as-grown ZnO nanowires with ~30−70 nm diameter and ~5−7 μm length. Inset: a selected area electron diffraction pattern of a single-crystalline nanowire. (**b**) HRTEM images of a ZnO nanowire indicating the spacing of ~2.6 Å between two crystalline planes along [0001] growth direction. Cross-sectional SEM images of a portion of (**c**) a porous PDMS film and (**d**) a nanocomposite porous PDMS film with randomly distributed ZnO nanowires. Note that the arrow marks indicate the existence of nanowires. (**e**) Photo images of fabricated flexible ZnO nanowire–porous PDMS capacitive pressure sensors: a single cell (top) and a 4 × 4 multipixel array (bottom).

**Figure 3 nanomaterials-12-00256-f003:**
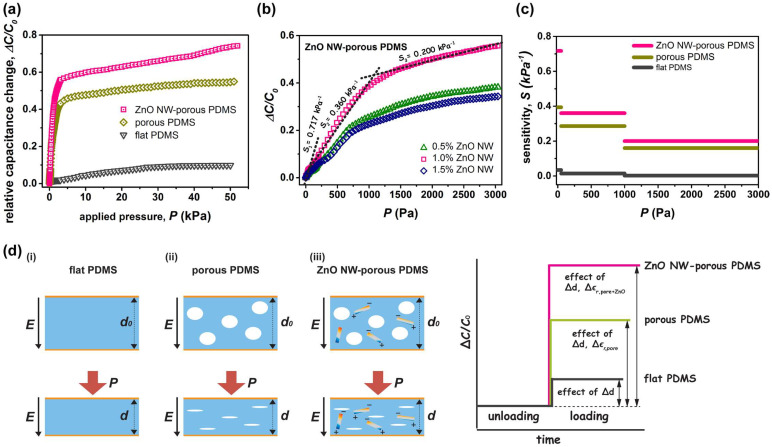
(**a**) Measured relative capacitance change (Δ*C/C*_0_) as a function of the applied pressure (*P*) for the capacitive pressure sensors with different types of dielectric layers: flat PDMS, porous PDMS, and ZnO nanowire–porous PDMS (sensing area: 10 mm × 10 mm). (**b**) Pressure-response plots for these nanocomposite sensors with varying ZnO nanowire loading. (**c**) Comparison of the sensitivity of capacitive pressure sensors at different applied pressure ranges. (**d**) Proposed sensing mechanisms with graphical capacitance change for the capacitors containing (i) flat PDMS, (ii) porous PDMS, and (iii) ZnO nanowire–porous PDMS.

**Figure 4 nanomaterials-12-00256-f004:**
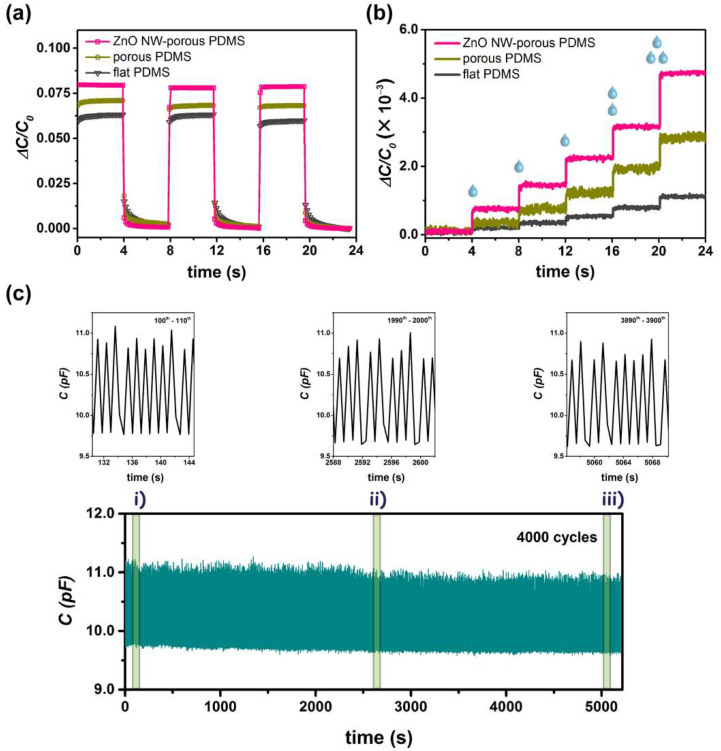
(**a**) Time-resolved capacitive response of nanocomposite capacitive pressure sensors based on flat PDMS, porous PDMS, and ZnO nanowire–porous PDMS, respectively. (**b**) Limit of detection test by means of the sequential detection of water droplets. (**c**) Stability test of a ZnO nanowire–porous PDMS device for 4000 cycles at 300 Pa; magnified curves of (i) 100−110, (ii) 1900−2000, and (iii) 3890−3900 cycles, respectively.

**Figure 5 nanomaterials-12-00256-f005:**
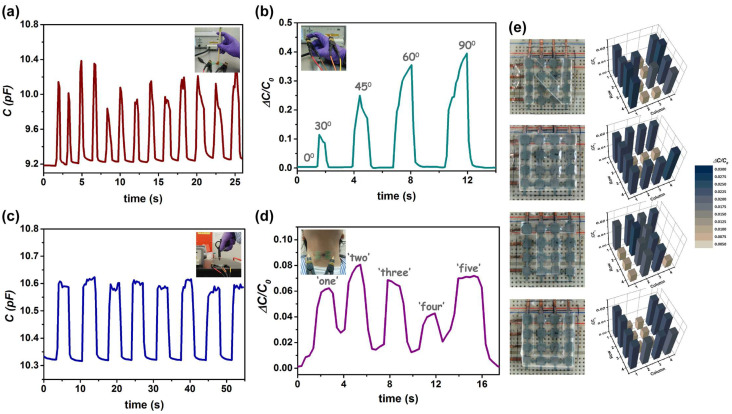
Real-time capacitance variations of the sensor in response to (**a**) calligraphy writing, (**b**) index finger straightening and bending, (**c**) air streaming, and (**d**) vocal speaking. (**e**) Photographs of a working pressure sensor array and their spatial pressure responses to the letters N, C, T, and U.

## Data Availability

Data is contained within the article and the Appendix A.

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
