# Peer review of "Porous Polydimethylsiloxane Elastomer Hybrid with Zinc Oxide Nanowire for Wearable, Wide-Range, and Low Detection Limit Capacitive Pressure Sensor"

_nanomaterials, 2022, doi:10.3390/nano12020256_

Round 1

Reviewer 1 Report

  1. The authors should provide a table comparing all the features of recently developed capacitive pressures sensors. It will make readers know what the advantages of this reported approach.
  2. Does porogen portion influence the porosity? If yes, how.
  3. Ultrasoft capacitive sensors have been demonstrated to monitor finger touch, e.g., Science 370.6519 (2020): 966-970. Please include the reference and briefly discuss the difference.
  4. Figure 2 should be rearranged. c and d do not have much difference.
  5. Figure 6 could be moved to supporting materials.

Reviewer 2 Report

This manucript is on the preparation of PDMS/ZnO films for a flexibe capacitive pressure sensor. The results are interesting, and the LOD is low with an ultrafine resolution. Thus, I recommend this manuscript for publication in Nanomaterials after addressing the following questions.

  1. Some evidences should be given to demonstrate the uniform distribution of ZnO in the nanocomposite film, e.g. SEM image of cross-section view, Raman or FTIR spectrum.
  2. How to define ΔC?
  3. On page 5, line 192, the sentence “This was probably qattributed to the elastic modulus of the nanocomposite…”. How much is the elastic modulus of the flat PDMS, the porous PDMS and the PDMS/ZnO film, respectively?
  4. What does the caption of Fig. 3a mean?
  5. According to Fig. 3a, how about the sensitivities of the three materials when the applied presure was larger than 3000 Pa?
  6. Comparing Fig. 4a and Fig. 3b, the relative capacitance change of the PDMS/ZnO film was different under the same pressure (300 Pa), why?

Reviewer 3 Report

The manuscript by Hsieh et al. reports the contribution of ZnO nanowires to porous PDMS in terms of improving pressure sensitivity. In particular, the sensitivity and endurance of the proposed pressure sensors have been validated in quantification analysis and detailed applications.

There should be a major revision before publication. There are still some unsound conclusions in this work that should be improved in the revised version, and more details should be provided. Comments are as follows:

  1. The reproducibility of the PDMS membrane fabrication process is an issue. Porous in PDMS exhibit a wide range of size and distribution in Fig. 2(c-d). Compressive properties of PDMS membranes are primarily due to the porous structure, which appears to be generated randomly and unpredictably.

  1. Line 122: “subsequent backing at 70 ℃ for 24 h” for the PDMS. During the solidification of PDMS, the baking temperature and time can affect the size and distribution of porous. Therefore, please explain why you chose this temperature and time, and what about the other parameters?

  1. The details about fabricating flat PDMS sensors should be provided in the section of 2.2.

  1. Lin 124: “PDMS film (thickness of ~200 μm)”. This work focused on the compressibility and dielectric properties of porous PDMS membrane, therefore, the thickness is important. How to determine the effect of the thickness of PDMS film? How about 100 μm or 400 μm?

  1. Line 220. “In the regime of 1000 – 3000 Pa, the ZnO nanowire-porous PDMS….”. The proposed sensor only outperformed flat PDMS sensors (100 times better) within this pressure range, and authors should provide the comparison results to porous PDMS sensors. Additionally, in this pressure range, the conclusion is incorrect; the value of the porous PDMS sensors is not saturated, and only the flat PDMS sensors are saturated.

  1. Line 222, “excessive content of ZnO nanowire loading adversely affected…”. So please explain this situation. Does excessive content of ZnO nanowires influence the formation of the porous structure in a PDMS film? If yes, please provide cross-section SEM images as evidence.

  1. Line 245. “Embed air pores make the ZnO nanowire-porous PDMS more susceptible to deformability”. So what is the difference between the porous PDMS and ZnO nanowire-porous PDMS in terms of deformability? Authors should clarify the function of the ZnO nanowires in this work. Does the ZnO nanowire change the dielectric properties of the PDMS membrane or soften the porous membrane?

  1. Line 283. Authors should repeat similar endurance tests on the porous PDMS sensors. And then, the contribution of ZnO nanowires on the duration of the sensors should be provided.

Round 2

Reviewer 2 Report

The manuscript can be accepted at the current state.

Author Response

The authors thank the referee for the excellent comment and those constructive feedbacks.

Reviewer 3 Report

Author addressed most of my concerns, however, there are still a few points should be revised

All the data/graphs used for the answering the comments of reviewer must be included into the supplementary material and cited in the main text.

Discussion of using other material should be added, PDMS is not chemical/physical stable, and easy for aging. Such as using flexible glass sheet could improve the durability and stability of the sensor.

Porous structure is kind of random arrangement of nano holes, the size, distribution and position of the holes slightly influence on the sensor performance, what about to arrange the position, size and distribution of the holes (such as by using femtosecond laser)? That is interesting for audience who want to further improve the potential of the sensors.

Author Response

Dear Referee

Thank you again for these constructive feedbacks and for recommending it for publications. Enclosed are our point by point responses to the comments. Supplementary Information is also attached separately.

COMMENT #1

All the data/graphs used for the answering the comments of reviewer must be included into the supplementary material and cited in the main text.

  • Authors’ Response to Comment #1:

The authors thank the referee for the suggestion. We have included all those data/graphs into the supplementary material, which has been cited in the main text accordingly.  

  • Modification to Manuscript regarding Comments #1:

Changes to manuscript and supplementary information have been made based on the above response.

COMMENT #2

Discussion of using other material should be added, PDMS is not chemical/physical stable, and easy for aging. Such as using flexible glass sheet could improve the durability and stability of the sensor.

  • Authors’ Response to Comment #2:

We thank for the comment. We have made a discussion, regarding the choice of materials, issue of porous structure formation, and future work in the manuscript.

  • Modification to Manuscript regarding Comments #2:

Changes to manuscript (paragraph 2, page 18) have been amended based on the above response.

COMMENT #3

Porous structure is kind of random arrangement of nano holes, the size, distribution and position of the holes slightly influence on the sensor performance, what about to arrange the position, size and distribution of the holes (such as by using femtosecond laser)? That is interesting for audience who want to further improve the potential of the sensors.

  • Authors’ Response to Comment #3:

The authors thank for the insightful perspective. We have made a discussion, regarding the choice of materials, issue of porous structure formation, and future work in the manuscript.

  • Modification to Manuscript regarding Comments #3:

Changes to manuscript (paragraph 2, page 18) have been amended based on the above response.

Thank you very much for your consideration.